# Ethogram-based Personalization of Human Activity and Agility from Radar $\mu$D Signatures

Emre Kurtoğlu[1], Sultanus Salehin[1], Moeness G. Amin[2], Irene P. Kan[3], Michelle A. McKay[4], Sevgi Z. Gurbuz[1]

[1]*Dept. of Electrical and Computer Engineering, The University of Alabama*
[2]*Dept. of Electrical and Computer Engineering, Villanova University*
[3]*Dept. of Psychological and Brain Sciences, Villanova University*
[4]*College of Nursing, Villanova University*

*Abstract*—**Radar has garnered great interest for remote health monitoring due to its ambient operation, effectiveness in the dark, and inability to make visual recordings of private scenes/faces. However, the current state-of-the-art in human activity recognition (HAR) focuses on the classification of persistent gaits, such as walking, and ignores the transitions between activities. The characterization of a person's ability to transition between postural states is highly individual and influenced by the person's physical and mental health. This paper presents a personalized, ethogram-based approach to HAR, which jointly characterizes the agility of transitions in addition to activity classification. We develop a multi-input multi-task learning (MIMTL) approach to simultaneously classify both human activity and agility. Our proposed approach yields accuracies of over 98% and 90% for the joint characterization tasks. Various interventions affecting gait are applied to show how the proposed approach can lead to agility-based detection of changes in gait.**

*Index Terms*—**Radar, human activity recognition, mmWave, remote health monitoring, $\mu$D.**

## I. INTRODUCTION

The world population is rapidly aging, with 2.1 billion adults 60 years of age and over expected globally by the year 2050 [1]. Mobility limitations currently affect one-third of adults 70 years of age and older, and most over 85 years of age are associated with functional decline, an increased risk for falls, and decreased quality of life [2], [3]. As the population grows, the number of older adults with mobility limitations will increase. Age-related changes (decreased muscle strength, alterations in joint flexibility and proprioception, sensory decline, reduced reaction time) affect posture and balance, causing the most common daily movements of sitting, standing, and walking more demanding and difficult [4], [5]. The ability to transition between the postural states of sitting and standing, known as postural transition, is an important indicator of mobility and functional independence [5]. Falls occur frequently during walking as well as postural transitions between sit-to-stand or stand-to-sit movements [6]. Therefore, measuring and quantifying postural transition duration (TD) is relevant for clinical assessment of older adults to identify changes in function, prevent falls, and develop targeted interventions to improve mobility and maintain functional independence.

The ability to transition between static postural states is individual and influenced by the onset of acute disease, chronic disease, or changes to a person's mental or physical health.

Among healthy older adults, there is a range of postural TD times that can be categorized as slow, normal, and fast [7]. Frail older adults [7], older adults who have suffered a stroke [4], persons with Parkinson's disease [8], older adults after total knee arthroplasty (TKA) surgery [9], and older adults who have experienced a fall have significantly increased postural TD times when compared to healthy older adults [10], [11]. By observing variations in TD between postural states, and the wide range of potential postural movements, an inclusive ethogram that characterizes these states and movements can be used to detect slight changes in transitions and postural states, and these capabilities can allow for rapid and accurately identify potential risks for functional decline, falls, and the associated negative outcomes for older adults.

While there exists people with a wide range of ages, mobility profiles and treatment histories, current state-of-the-art human activity recognition (HAR) methods often focus on building a generalizable recognition model for everyone disregarding their agility conditions. Models are trained by recording data from a large number of participants, whose ambulation varies in style and speed. However, this type of generic training precludes the detection and characterization of personalized ambulation, causing systems to under-perform when they are encountered with an unseen or abnormal way of performing an activity. Therefore, it is important for the system to characterize the agility (i.e., mobility) profile of a person along with the activity. Any unusual pattern or abnormality in the personalized ethogram can be an indicator of certain health issues or improvements in a health condition, depending on the direction of the change.

In HAR applications, various sensor modalities have been employed such as cameras [12], wearables [13], inertial measurement units (IMUs) [14] and radio frequency (RF) sensors [15], [16]. Radars have proven to be a promising sensor technology for health applications such as vital sign monitoring [17], fall detection [18] and fall risk assessment [19]. Radars present distinct advantages over other sensor modalities for indoor monitoring applications such as high range and velocity resolution, low cost, small package size, and protection of privacy through their inability to record visual imagery of the face or background. The purpose of this paper is not to cast radar as a superior sensor to other sensing modalities, including wearables, but rather demonstrate the

utility of a single RF sensor for human motion classification with agility determination. A network of radar sensors would offer different aspect angles to the subject and has been generally shown to outperform a single radar unit [20]. The findings of the radar sensor or network can also be fused with information gleaned from other deployed sensors, if available.

In a smaller-scale, preliminary study [21], it has been shown that the recognition of activities of daily living (ADL) could be improved by jointly classifying activity and agility. This work further develops radar-based algorithms for agility characterization by using joint optimization of both tasks to characterize and recognize changes in agility incurred due to interventions - the use of knee and leg braces or stiffening of the back as a proxy for emulating minute gait aberrations that can affect agility. In particular, we characterize agility in addition to activity as properties of a person's individualized mobility ethogram and show that the proposed multi-input multi-task learning (MIMTL) network can provide an agility score with which agility aberrations due to interventions can be detected. In this way, we show that deviations in agility could be used as a basis for radar-based gait abnormality detection.

This paper is organized as follows. In Section II, the ethogram as a model for mobility and the inclusion of an agility score as part of its quantification has been described. Next, we describe in Section III the datasets collected and analyzed, as well as the radar data pre-processing pipeline. Our proposed approach for agility characterization under nominal conditions and with interventions are described in Sections IV and V, respectively. Finally, we conclude in Section VI.

## II. ETHOGRAMS AS A MODEL FOR MOBILITY

Briefly, an "ethogram" refers to a catalog of behaviors that are specific to the species, and it represents a directory of discrete acts within a larger context [22]. To generate this inventory of behaviors, researchers would observe natural behaviors over a period of time (e.g., 15 minutes) and systematically document all the discrete and repeated behaviors that may occur (e.g., feeding, grooming). Across observations of multiple members of the same species, a directory of common behaviors for the species is established.

This approach has its origin in ethology and is advantageous for several reasons. First, a catalog of observable behaviors that includes objectively clear descriptions can provide some degree of standardization across research labs, which promotes transparency and consistency in analyses. Second, this inventory can be used to characterize both inter-individual and intra-individual differences. Take grooming behavior as an example. While all macaque monkeys engage in grooming behavior, the proportion of time each member spends on this activity varies (e.g., adult females spend more time grooming than adult males) [23]. Furthermore, variations within the same individual can also be captured easily. For example, the proportion of time spent on each daily activity may vary as a function of internal changes (e.g., an individual's physical state) and/or external changes (e.g., seasonal variations). Third, this inventory can be adapted over time to suit the needs of

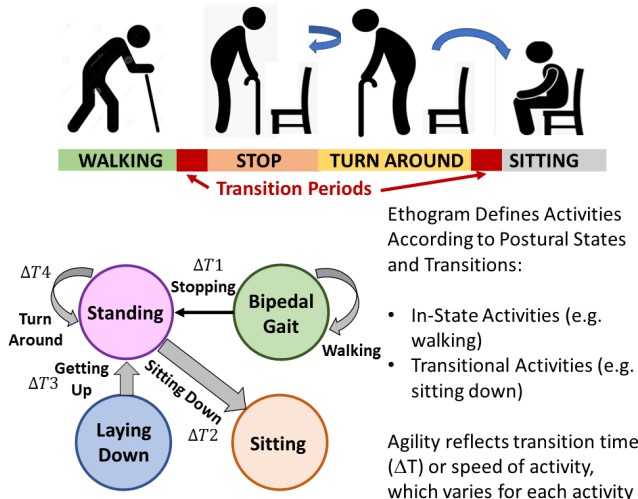

Fig. 1: Example ethogram: activities and agility.

the research question. For example, a researcher may opt to subdivide broader behaviors (e.g., foraging) into more fine-grained measures (e.g., foraging for leaves vs fruits) to better characterize the full complement of behaviors.

Ethogram-based characterization of observable behaviors has extended beyond the study of non-human animals [24]. For example, it has been applied successfully to surgeons' behaviors in operating room [25], patients' behaviors in the psychiatrist's office [26], and infants' distress response to pain [27]. With regards to activities of daily living–consider the act of getting up from a chair and walking to the door. Within this sequence, there are distinct components that can be easily identified, such as the transition from sitting to standing. While the components involved may be universal across humans, the nuance of each component, such as speed and trajectory, is idiosyncratic for each individual.

From a human activity point of view, human ethogram can be categorized into two types of motions: translational and in-place [28]. While the translational motions refer to displacement in gross body movements such as walking, crawling, and limping, in-place motions are mostly associated with activities that do not have a significant change in a person's distance to a reference point, such as sitting down, standing up, or turning around. Range information provided by the radar sensors enables the differentiation of transitional activities from in-place motions [29].

Ethograms provide a way to represent the mobility-constraints of sequences of activities. For instance, falling cannot be followed by walking without standing first. Or, sitting should always be preceded by a standing position. The sequence of activity is often health-related. For example, while a young, healthy person may be able to sit down rapidly, an older adult with fall risk may first need to stop, slowly turn around, and then sit, as depicted in Figure 1. This gives rise to a person-specific, health-correlated ethogram, characterized not just by the mobility-based possible postural transitions, but also by agility - the typical time it takes to transition or the

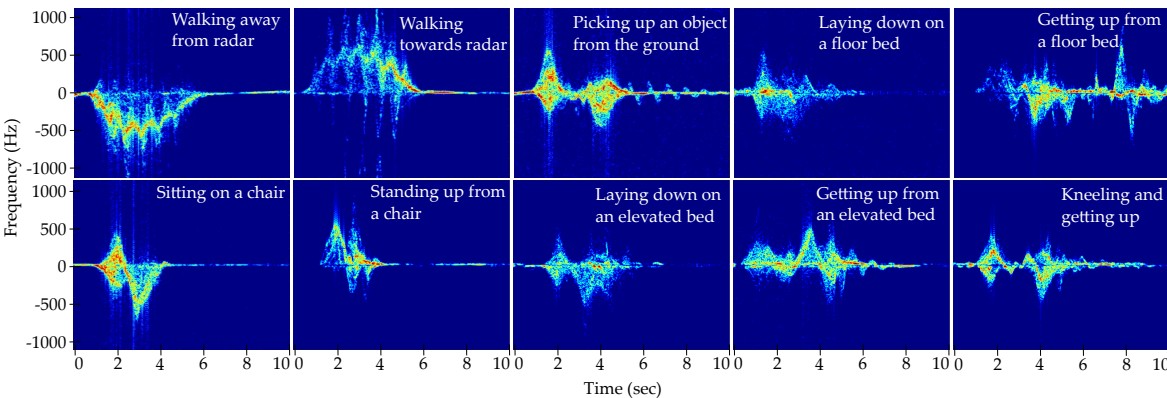

Fig. 2: μD spectrograms of different activities.

rate at which an activity is conducted.

This work contributes to the quantitative development of the ethogram for human activity recognition by adding the concept of agility to characterize transitions between states and shows that joint consideration of agility is essential, not only for activity recognition but also for the detection of gait abnormalities (in this paper, induced through interventions, but more generally, are the result of adverse health conditions).

## III. EXPERIMENTS AND DATA

### A. Interventions and Personas

In this work, we defined nine different mobility personas, motivated by studies in the literature, which document the resulting speeds and impact on agility of various health conditions. In associating a Persona with a mobility-impacting health issue, several metrics were considered, such as average walking speed and completion of standing-up and sitting-down activities. Using interventions, such as the walking boot, knee brace, or keeping the back stiff, role-playing was utilized. For example, when using walking boots, the actor mimicked the walking pattern of someone recovering from a broken foot by not fully putting their weight on that leg and dragging their foot at times. Thus, while associated conditions may impact ambulation, it is important to note that such a person may still be able to sit down or lay down without their agility in those activities being significantly affected. On the other hand, knee braces and stiff back interventions would affect the agility of sitting, standing up, and laying down. This underscores the need to jointly consider activity and agility when seeking to detect gait abnormality. The nine different Personas defined and their motivating conditions are listed as follows:

1) **Slow, Normal Movement** - emulating a healthy older adult with no significant physical or mental ailments [4]
2) **Nominal, Normal Movement** - emulating a healthy adult with no history of injury [9]
3) **Fast, Normal Movement** - emulating a healthy, young adult with no health issues
4) **Slow, with Walker Boot** - emulating a potential faller, who may have fallen within the past 5 years, and who can only move extremely slowly with some limp [10]

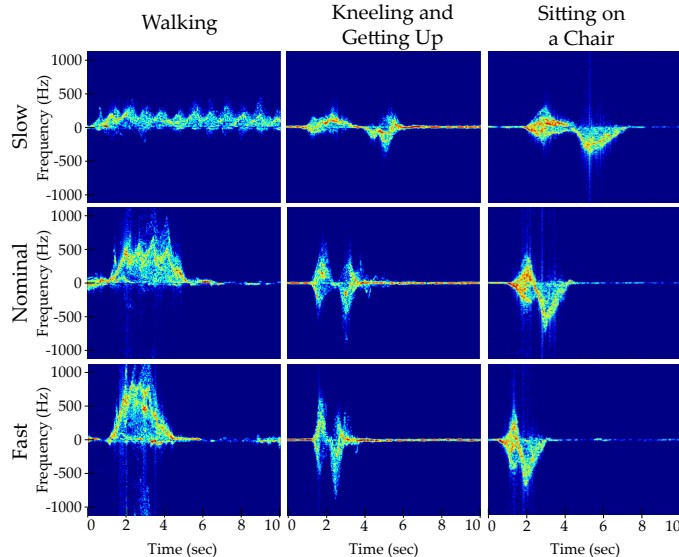

Fig. 3: μD spectrograms of activities for varying speeds.

5) **Nominal, with Walker Boot** - emulating a frail older adult, who may have low muscular strength [4]
6) **Slow, with Knee Brace** - emulating a total knee arthroplasty (TKA) patient [9]
7) **Nominal, with Knee Brace** - emulating an adult with knee osteorthritis, who may have pain, activity limitations, or self-reported knee instability [30]
8) **Slow, with Stiff Back** - emulating a frail older adult with potential fall risk [5]
9) **Nominal, with Stiff Back** - emulating a post-stoke recovery patient, who has some impairments [7]

### B. Radar and Signal Processing

Data was acquired using both a radar and an RGB camera, colocated with the radar, to aid in ground truth labeling. The radar used was a Texas Instrument's AWR2243BOOST frequency-modulated continuous wave (FMCW) multiple-input multiple-output (MIMO) radar coupled with DCA1000EVM data capture card for raw I&Q data streaming. This radar operates at 77 GHz with a 4 GHz bandwidth, enabling a 3.75 cm range resolution. The radar's antenna array consists of 3 transmitter (TX) and 4 receiver

(RX) antennas with the beamwidths of ± 70° in azimuth and ± 15° in elevation. Both camera and radar were placed 0.9m above the ground, and all the activities were performed in the direct line of sight of the sensors.

The raw radar I/Q data time stream was processed by first forming a 3D array of fast-time samples × slow-time samples × channels. Here, fast-time refers to the ADC sample interval, while slow-time refers to the pulse number and channels refer to the TX-RX antenna pairs of the radar. The micro-Doppler ($\mu$D) signature of the data is computed by first applying a Fast Fourier Transform (FFT) to each chirp. This results in a heat map known as the range profile (RP). Upon range detection, the $\mu$D signature is computed as the square modulus of the Short-Time Fourier Transform (STFT), also known as a spectrogram. An example $\mu$D spectrogram for the ten different activities recorded is depicted in Figure 2.

*C. Datasets*

In this work, two datasets are acquired to evaluate the proposed approach: one where no interventions are used (Personas #1-3), and one where interventions of a walker boot, knee brace and stiff back are employed (Personas #4-9).

*1) No Intervention Dataset:* Ten activities, including

- **WLKT**: Walking towards radar
- **WLKA**: Walking away from radar
- **LAYB**: Laying down on an elevated bed
- **GETB**: Getting up from an elevated bed
- **LAYF**: Laying down on a floor bed
- **GETF**: Getting up from a floor bed
- **SITC**: Sitting down on a chair
- **STDC**: Standing up from a chair
- **PICK**: Picking up an object from the ground
- **KNEE**: Kneeling and getting up

Each activity was enacted according to three unique agility scores - *slow*, *nominal*, and *fast* - to represent Personas #1-3. Slow activities are executed in a manner typical of older adults or individuals with limited mobility [4]. A person may also move slowly when preoccupied with tasks such as reading,

carrying, or gripping objects. Nominal activities are performed at an average pace and are typical of healthy adults [9]. Finally, fast activities are typically associated with healthy, younger individuals [31]. Based on these descriptions, participants acted with role-play the ten daily activities at different speeds.

Six participants of varying ages and genders took part in the study, and each recording lasted 10 seconds. A total of 900 samples were acquired, which were then divided into 70% training and 30% testing splits.

*2) Sequential Activities with Interventions:* This dataset captures sequences of daily activities, where participants were instructed to role-play according to the Personas with interventions (#4-9). The use of the interventions (walker boots, knee braces, and a stiff back) result in small aberrations in gait and for different activities will affect the measured agility score differently. This dataset consists of six activities: Walking towards the radar (WLKT), laying down on a bed (LAYB), getting up from a bed (GETB), sitting on a chair (SITC), standing up from a chair (STDC) and picking up an object from the ground (PICK). These activities are conducted sequentially, as in a real-life use case, where participants were instructed to conduct the six activities in the following order with a short pause between each activity:

WLKT → SITC → STDC → PICK → LAYB → GETB

The total data recording lasted for 70s for Personas #4-6-8 and 50s for Personas #5-8-9. In total, 103 recordings of the sequence were acquired from six participants.

*D. Effect of Agility on RF Data*

Because the $\mu$D spectrogram represents velocity versus time, agility has the effect of compressing or expanding the signature in time, while also potentially increasing or decreasing the peak/minimum velocities of the envelopes. In other words, a more agile articulation would occur faster, over a shorter duration of time. This effect can be observed in the example spectrograms shown in Figure 3 for various agility levels and in the distributions of the average maximum velocities plotted in Figure 4. Using a percentile technique to extract the upper and lower envelopes of the $\mu$D signatures, we found

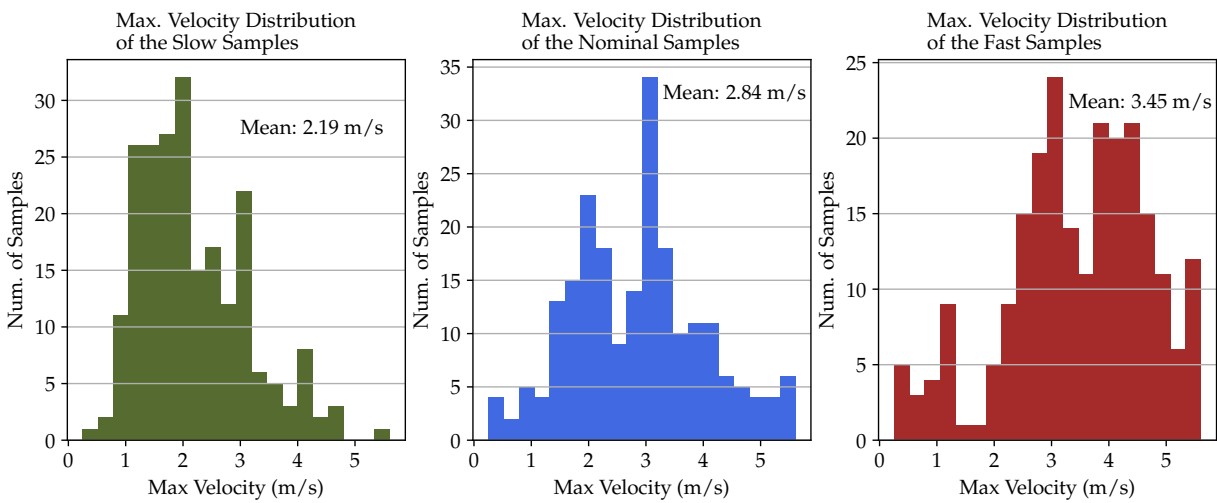

Fig. 4: Maximum velocity distributions of different agility scores.

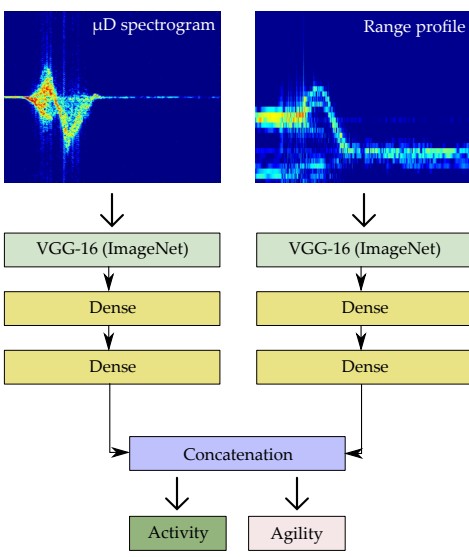

Fig. 5: Multi-input multi-task learning network for joint activity and agility score prediction.

that *slow* samples have the lowest maximum velocity amongst the agility levels with an average of 2.19 m/s. *Nominal* and *fast* samples, on the hand, have average maximum velocities of 2.84 m/s and 3.45 m/s, respectively. Note that some overlap in the agility distributions is normal and expected given that the motions are articulated via role-playing and such variations are to be expected in real scenarios.

## IV. Joint Agility and Activity Recognition

The consideration of agility when classifying activity is important because the activity class can otherwise be confused. For instance, one can sit on the floor from a standing position; but if this were to happen rapidly, we may better conclude that the person had fallen. To enable joint consideration of agility and activity in the classification process, we thus propose the use of MIMTL, where not just the $\mu$D signature but also the range profile is utilized at the input, and the network is trained to optimize for ascertaining both agility and activity recognition tasks. The proposed MIMTL network is shown in Figure 5, where VGG-16 pre-trained with ImageNet dataset is employed as a backbone network to provide a more suitable initialization of the network under limited data.

To demonstrate the merits of the proposed approach, we compare MIMTL for joint agility+activity classification with networks that 1) classify only activity, 2) classify only agility, and 3) classify all combinations of activity and agility scores with a single task network. To ensure a fair comparison, we used the same backbone network (VGG-16) for feature extraction across all methods. Note that while the activity-only network has 10 neurons and the agility-only network has 3 neurons in the output layer, the combined activity+agility network requires 30 neurons (10 activities x 3 agilities) for joint prediction. In contrast, the multi-task learning approach features two separate output layers, with 10 neurons for activity prediction and 3 neurons for agility prediction. The combined loss for these two tasks is $\mathcal{L}_{\text{total}} = \lambda \mathcal{L}_{\text{act}} + \gamma \mathcal{L}_{\text{ag}}$,

TABLE I: Agility and Activity Recognition Performance.

| Recognition Method | Number of Output Classes | Acc. (%) Activity | Acc. (%) Agility |
|---|---|---|---|
| Activity Only | 10 | 95.9 | - |
| Agility Only | 3 | - | 89.7 |
| Joint Act. & Ag. | 30 | 89.7 | 91.9 |
| Multi-task Act. & Ag. | 10 & 3 | 95.2 | 90.8 |

where $\mathcal{L}_{\text{act}}$ and $\mathcal{L}_{\text{ag}}$ are the categorical cross-entropy losses for activity and agility recognition tasks, and $\lambda$ and $\gamma$ are the weights of the corresponding losses. Both $\lambda$ and $\gamma$ are set to 1 to avoid any bias towards a particular task.

Table I presents the classification results for the four described methods. While the activity-only and agility-only networks achieve high classification accuracy rates of 95.9% and 89.7%, respectively, they are designed for single tasks and require separate networks, which doubles the memory and processing power requirements. Although the joint activity+agility network has an overall testing accuracy for 30 classes of 82.3%, it achieves task-wise accuracies of 89.7% and 91.9%. Our proposed MIMTL network improves the activity recognition accuracy relative to joint prediction by over 5%, yielding an accuracy of 95.2% while maintaining comparable agility classification at 90.8%.

The impact of utilizing multiple inputs (both range and $\mu$D) as well as multi-task optimization versus joint (combined) classification can be seen in Figure 6. Note that when range information is incorporated into the network architecture, the activity recognition accuracy increases from 89.7% to 95.6% and from 95.2% to 98.2% for joint and multi-task networks, respectively. Moreover, using multi-task learning instead of joint learning of the tasks improves the accuracy from 89.7% to 95.2% and 95.6% to 98.2% for single and multi-input networks, respectively. For the agility recognition task, there is only 1% performance drop when the model is changed from joint to multi-task learning for single input case, and a slight increase from 88.2% to 90% for multi-input case.

## V. Characterization of Gait Changes

In this section, we examine how we can use the proposed MIMTL network as a tool for characterizing gait changes induced through the use of interventions. The personas with interventions (#4-9) induce slight gait changes in the participants. More broadly, many health conditions can cause a variety of gait abnormalities, the detection of which can be useful in remote health monitoring. Not all health conditions affect agility across every activity.

Similarly, each intervention has a different impact on the participant's mobility. For example, a person with a knee brace may need to hold onto furniture for support before attempting to sit or lie down, resulting in a slower than normal movement. But when lying down, they might suddenly drop to the bed, unable to control their descent due to the knee condition, resulting in a faster than normal movement. Conversely, a

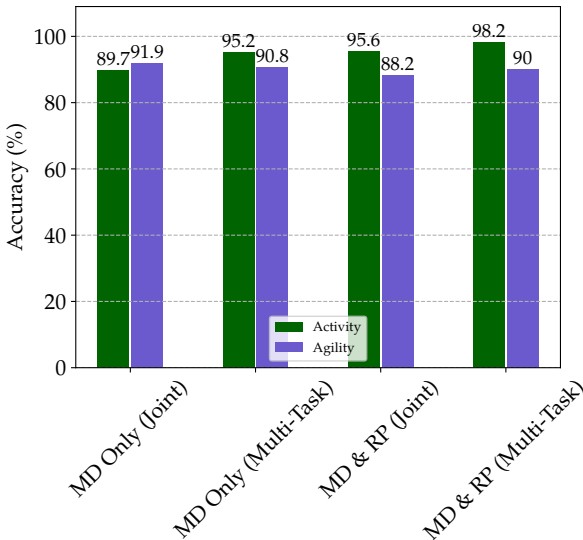

Fig. 6: Comparison of joint and multi-task estimation results for single and multi-input networks.

person with a stiff back would carefully sit on the bed smoothly before lying down. Or, a person with minor ankle injuries may not be significantly affected when lying down but might exhibit increased sensitivity and caution while walking or kneeling. Thus, as illustrated by these examples, specific interventions can lead to variation in agility for different activities, and observing these changes can provide valuable insights into a person's agility, and, consequently, health.

The gait changes induced through the interventions can be visibly seen in the $\mu$D signatures of the activities. Figure 7 compares the $\mu$D spectrograms for normal walking with that of walking with the use of a knee brace, walker book or stiff back. Despite similar average velocities, sharper sawtooth-like transitions occur with knee braces and walker boots due to interrupted movement of the injured foot/leg. Conversely, individuals with stiff backs walk more smoothly with fewer abrupt movements and a fixed posture. Similar patterns are observable for other activities, demonstrating the radar's ability to capture detailed features of human movement.

### A. Classification Robustness of Data with Interventions

An important requirement for any HAR system is that it should be able to robustly classify activities, despite any potential gait changes. To evaluate the activity recognition robustness of the proposed MIMTL approach, the model trained with only data acquired from Personas #1-3, which reflect normal, unrestricted human motion, is tested on the data of Personas #4-9, where participants utilized interventions. Figure 8 presents the resulting confusion matrix, showing an overall accuracy of 96.3%. There was only one misprediction for LAYB and WLKT activities, and at most four mispredictions for the STDC activity out of 75 samples. This demonstrates the generalization capability of the proposed method when the system is tested on HAR data exhibiting gait changes.

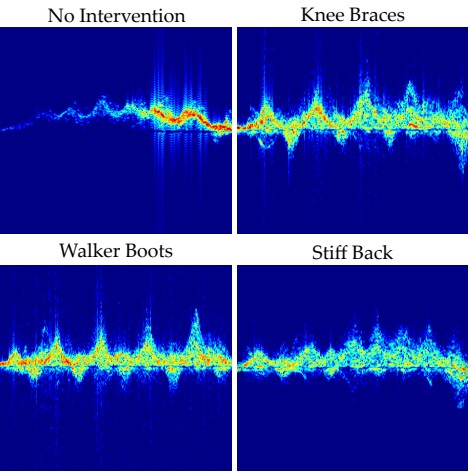

Fig. 7: WLKT $\mu$D signatures under different interventions.

Additionally, we evaluate the performance of the model for each intervention separately. Figure 9 presents the activity recognition results of the MIMTL network for different interventions. Personas with the same type of interventions are merged for brevity. It can be seen that all the activities performed with walking boots are classified correctly. While the Personas with knee braces yielded a comparable accuracy to the overall performance, it is observed that majority of the mispredictions in the overall results were stemming from the Personas with stiff backs. Amongst the activities for stiff back Personas, STDC, GETB and PICK classes yielded the lowest accuracies of 84.2%, 87.5% and 86.7%, respectively. The remaining SITC, LAYB and WLKT classes yielded accuracies of 94.7%, 93.8% and 95%, respectively. Thus, we can see that the greatest gait changes are induced through keeping the back stiff, while using walking boots has the least impact on gait.

### B. Agility Characterization and Detection of Gait Changes

Because different interventions vary in which activities are affected, observations of a person's sequential movements result in not just a single scalar agility score, but in a vector of agility scores, where each activity can have its own associated agility score. To illustrate this point, Figure 10

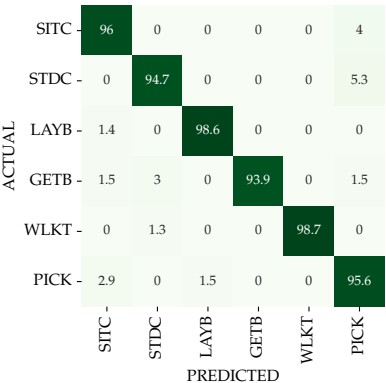

Fig. 8: Confusion matrix for testing MIMTL network for Personas with interventions ([4-9]).

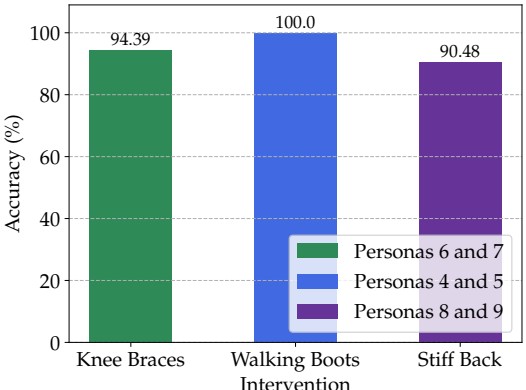

Fig. 9: MIMTL performance for different interventions.

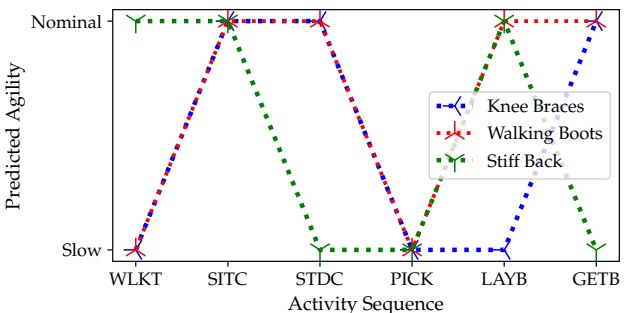

Fig. 10: Predicted agility scores of different interventions for an activity sequence.

shows the agility scores predicted for each activity in the sequential data described in Section III-C2 across different interventions. As may be observed from the sequential agility vector plotted, both knee braces and walking boots cause slower walking compared to normal motion. However, sitting down and standing up are less affected. While individuals with walking boots lay down at a regular pace, those with knee braces do so more slowly due to their inability to bend their knees. Both can get up from bed at a normal pace. A person with a stiff back walks and sits without difficulty but struggles with standing up, bending down to pick up objects, and getting up from bed. The agility score for sitting down is less affected while picking up objects is impacted in all cases. While these results are drawn from a single activity sequence from each intervention type, similar patterns are observed in other sequences, demonstrating the radar's capability to recognize the changes in the agility profile of a person.

Characterizing agility alongside recognized activities provides unique insights into a person's mobility profile. Agility scores vary across activities, and changes in agility can indicate shifts in a person's ethogram and activities. Detecting such changes is crucial for health monitoring. For example, walking slower than usual can indicate cardiovascular diseases [32] and potentially lead to severe incidents if not detected early. Conversely, an increase in walking speed may result

from regular exercise and improved mobility. Monitoring and analyzing these changes over time offer valuable information about a person's health trajectory, enabling early diagnosis.

To detect gait changes based on agility, we augment the MIMTL network with an additional binary output layer. In this study, a gait change is assumed to only occur with interventions (i.e., walking boots, knee braces, or stiff back). Thus, for Personas without interventions (#1-#3) the detection label is set to zero, while those for Personas with interventions (#4-#9), it is set to one. Data from Personas with interventions (#4-#9) are split equally for fine-tuning and testing. The MIMTL model is then fine-tuned with the new tasks.

Table II presents the abnormality detection results for different intervention types and activities. Table IIb shows that SITC and STDC have the lowest abnormality detection rates, as walking boots minimally affect sitting down or standing up. Conversely, GETB, LAYB, and PICK activities exhibit high (>90%) detection rates. Abnormalities with knee braces are also detected at high rates, except for WLKT, which has lower detection rates across all interventions. This variability may stem from differences in how participants role-play and perform activities based on their perception of what is natural.

## VI. CONCLUSION

This work presents a multi-task learning approach to jointly estimate human activity and the associated agility via RF sensing. We show that adding agility estimation as an auxiliary task to the network augments information and acts as a regularizer during model training. In addition, estimation of

| Activity | Detection Ratio (%) | Total Num. of Samples |
|----------|---------------------|------------------------|
| SITC | 88.9 | 9 |
| STDC | 100 | 8 |
| LAYB | 100 | 9 |
| GETB | 91.7 | 12 |
| WLKT | 50 | 12 |
| PICK | 100 | 11 |

(a) Knee Braces (Personas 6 & 7)

| Activity | Detection Ratio (%) | Total Num. of Samples |
|----------|---------------------|------------------------|
| SITC | 33.3 | 9 |
| STDC | 36.4 | 11 |
| LAYB | 87.5 | 8 |
| GETB | 100 | 11 |
| WLKT | 66.7 | 9 |
| PICK | 90.9 | 11 |

(b) Walking Boots (Personas 4 & 5)

| Activity | Detection Ratio (%) | Total Num. of Samples |
|----------|---------------------|------------------------|
| SITC | 100 | 9 |
| STDC | 77.8 | 9 |
| LAYB | 57.1 | 7 |
| GETB | 100 | 8 |
| WLKT | 50 | 10 |
| PICK | 100 | 6 |

(c) Stiff Back (Personas 8 & 9)

TABLE II: Abnormality detection results for different interventions.

agility helps to characterize the person's ethogram to understand unexpected behaviors or abnormalities which can be indicators of related health issues or improvements. Multi-task learning outperformed other single-task recognition networks and combined estimation approaches. Furthermore, we show that providing range as an additional input to the network enriches the feature space and yields better activity recognition performance while preserving agility score estimation accuracy. The resulting MIMTL detector enables consideration of both agility and activity, which can be critical in early warning for health conditions that may impact and change gait.

While the feasibility of the proposed approach is demonstrated for a limited number of interventions, the abnormality detection branch of the network is agnostic to the intervention type which makes it scalable to a broader range of interventions. A continuation of our work would require design and conducting an experimental study with objects from different age groups and health conditions which enables acquiring a diverse dataset. One can then proceed to build end-to-end remote activity and health monitoring applications. Finally, our work extends mobility ethograms to include agility characterization, advancing personalized radar-based HAR.

## ACKNOWLEDGMENT

This work was funded in part by NSF awards #223503 and #2238653. Research involving human subjects was carried out in accordance with UA Institutional Review Board (IRB) Protocols #18-06-1271 and #23-04-6553.

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
