# OpenReview forum: "Ethogram-based Personalization of Human Activity and Agility from Radar Micro-Doppler Signatures"
_IEEE.org/EMBS/BHI/2024/Conference — IEEE BHI'24_

### Official Review · Reviewer_ZnCq · 2024-07-31
**The work presented in this paper is in the scope of the conference and addresses a topic with potential in future applications for elderly people. It could benefit from some information about the work limitations and future work.**

**Overall Rating:** 6
**Confidence:** 2

**Other Quality Metrics:**

Clarity of writing: Good; Clinical Significance: Fair; Methodological Novelty: Fair; Experiments and Results: Good.

**Questions For The Authors:**

Although there is not much space left, the paper could benefit of adding some information about the limitations of the work done and future work.

**Strengths:**

The paper proposes a joint detection of agility and activity and obtains better results compared with individual parameters detection.

**Summary Of The Paper:**

This paper describes a solution based on Human Activity Recognition for measuring, quantifying and detecting changes in postural transitions in order to relate them with potential risks for functional decline and/or falls. The work uses data acquired with a radar and a RGB camera and shows a clear potential application for elderly people.

**Weaknesses:**

The paper describes the use of nine different personas and the data simulation for them; however, it is not clear how the proposed solution will still work when applied to real cases in the future.

---

> ### Author Rebuttal · Authors · 2024-09-03
>
> The authors would like to thank the reviewer for the constructive feedback and comments. We acknowledge that the current work is limited to the nine personas, and that validation in how these simulations reflect real world scenarios is required further validation. Nonetheless, this initial study demonstrated the efficacy of the proposed methodology for a smaller number of cases, and we plan to build upon these initial findings in future studies.
>      Our immediate plan is to design a joint experimental study to evaluate the proposed system under more realistic environments with people from different age groups with various health conditions, having real interventions. The planned study will be conducted in collaboration with two universities after the IRB approvals from both institutions. In addition, the abnormality detection branch of the network is agnostic to the intervention type which enables the proposed approach to be scaled to larger number of interventions. Therefore, we believe that the proposed approach can be extended to the real-world use cases when a sufficiently diverse dataset is acquired and followed by a rigorous testing.
>      We can also add to the conclusion section of the paper to briefly discuss the limitations of the proposed methodology and the future work.

---

### Official Review · Reviewer_Qweg · 2024-08-06
**Very interesting approach on task and abnormalities classification with radar data, with hard to contextualize results.**

**Overall Rating:** 7
**Confidence:** 4

**Other Quality Metrics:**

The clarity of the writing is generally clear, providing sufficient information to fully replicate the proposed work.
The study's clinical significance could be considerable, given the growing importance of real-time detection of abnormalities and pathological conditions in our increasingly aging society. The intrinsic ability of radar data to prevent subject identification by not capturing personal information is also a very appealing feature in today’s context.
The methodological novelty is suitable for publication at the BHI conference. The interesting approach of using an ethogram to better identify key activities and transitions is particularly noteworthy.
However, the results are somewhat difficult to interpret within the context of task and abnormality classification. The lack of comparison with similar works, or at least considerations regarding the advantages and potential impact of these results, is a shortcoming.

**Questions For The Authors:**

It would be beneficial to include an image illustrating the relative position of the radar in relation to the subject to enhance the clarity of Part B of Section III.

**Strengths:**

Although the data sample available is limited, the authors have effectively compensated for this by employing a task and agility classification approach that utilizes pretrained networks. Additionally, the analysis of the classifier's performance is well-designed and thoroughly documented.

**Summary Of The Paper:**

This study advances radar-based algorithms for agility characterization by jointly optimizing the tasks of characterizing and recognizing changes in agility due to interventions. The researchers define agility and activity as components of an individual's mobility ethogram and demonstrate that a multi-input multi-task learning (MIMTL) network can generate an agility score to detect aberrations caused by these interventions. The findings suggest that changes in agility could serve as a foundation for radar-based detection of gait abnormalities.

**Weaknesses:**

Although the work is remarkable, the results are somewhat difficult to situate within the current state of the art. The classification level achieved by the author is not superior to many other works focusing on task detection or abnormality detection using camera-based or wearable sensor approaches, for instance. However, there is a clear advantage in terms of privacy, minimal invasiveness, and other factors when using radar, which may compensate for lower performance. In my opinion, this aspect is not sufficiently described, making it challenging to fully understand the potential impact of the proposed work.

---

> ### Author Rebuttal · Authors · 2024-09-03
>
> The authors would like to thank the reviewer for the constructive feedback and comments. We agree that there are certain advantages and disadvantages of each sensor modality in remote health monitoring applications. Our intention in this work is not to suggest or recommend replacing other sensor modalities or to prove the superiority of radar against other sensors. Instead, we sought to demonstrate the efficacy of radars as a contactless device which can be used either as a standalone sensor platform or in conjunction with other sensors when these sensors become infeasible.  The latter includes cases such as insufficient light that prevents cameras from  effectively monitor the patient, and patient inconvenience or reluctance of wearing monitoring devices. Indeed, benchmarking the performance of different sensor modalities based on a commonly acquired dataset would be an invaluable study, but it is beyond the scope of this paper.
>      The radar was positioned 0.9m above the ground, and the participants performed all the activities in the direct line of sight of the radar within a 4m distance. Although there is not much space left for an additional figure, we can include this sentence in the dataset description section of the paper.

---

### Official Review · Reviewer_YebL · 2024-08-12
**Personalized Radar-Based Human Activity Recognition and Agility Assessment for Improved Health Monitoring**

**Overall Rating:** 6
**Confidence:** 1

**Other Quality Metrics:**

(a) Clarity of writing: fair
(b) Clinical Significance: good
(c) Methodological Novelty: good
(d) Experiments and Results: fair

**Questions For The Authors:**

1. What other interventions can be used to validate the effectiveness of MIMTL model in the study ?
2. How does the MIMTL model perform when applied to individuals with different mobility profiles, such as those with severe mobility impairments or neurological conditions ?

**Strengths:**

The paper presents a concept of integrating agility transitions into HAR, which addresses a critical gap in existing methods that often overlook transitions between activities.

**Summary Of The Paper:**

This paper introduces a novel approach to Human Activity Recognition (HAR) using radar technology, emphasizing the importance of characterizing not just persistent gaits but also the transitions between postural states. The authors propose a personalized and ethogram-based method that jointly classifies human activity and agility using a multi-input multi-task learning (MIMTL) framework. The approach demonstrates high accuracy in recognizing activities and detecting agility deviations caused by various interventions, making it a promising tool for remote health monitoring.

**Weaknesses:**

The interventions in the study, such as the use of knee and leg braces or stiffening of the back, may not fully represent the wide range of factors that could affect agility in real-world scenarios. It could benefit from more diverse interventions to validate the approach.

---

> ### Author Rebuttal · Authors · 2024-09-03
>
> The authors would like to thank the reviewer for the constructive feedback and comments. We acknowledge that the number of intervention types considered in this work is limited, and in real world scenarios, the number of possible intervention types are much larger. Other intervention types can include neck braces, arm braces, back braces, stiff neck or other limps, using cane, crutches or walker. While inclusion of all possible intervention types may not be feasible in terms of time and labor, the proposed approach can be scaled to other applications with larger datasets. In addition, the abnormality detection branch of the MIMTL network is agnostic to intervention type. This feature enables it to be easily adopted for other intervention types by fine-tuning the corresponding sub-network weights. We also intend to design and conduct an extended version of the study with people from different age groups and health conditions upon IRB approvals from both institutions where this study is to be conducted.
> We plan to add a brief discussion of the potential limitations (e.g., relatively small number of intervention types) and our future validation approaches to the conclusion section of the paper .

---

### Decision · Program_Chairs · 2024-09-23

Accept